# Understanding the Effects of Alignments between the Depth and Breadth of Cloud Computing Assimilation on Firm Performance: The Role of Organizational Agility

**Abul Khayer** [1],* [iD]**, Mohammad Tariqul Islam** [2] [iD] **and Yukun Bao** [3] [iD]

1   Department of International Business, University of Dhaka, Dhaka 1000, Bangladesh
2   Department of Management Information Systems, University of Dhaka, Dhaka 1000, Bangladesh
3   Centre for Modern Information Management, School of Management, Huazhong University of Science and Technology, Wuhan 430074, China
*   Correspondence: akhayer@du.ac.bd

**Abstract:** The main objective of this research is to generate insights about the effect of the depth and breadth of cloud computing assimilation on firm performance. The authors construct a research model based on several strands of theories to achieve the objective. This study considers two implementation alignment strategies: balanced fit and complementary fit, reflecting strategic choices made by organizations about the importance of the depth and breadth of cloud technology implementation. The model has been tested using a survey of Chinese businesses. The empirical study results show that cloud computing assimilation's depth and breadth, directly and indirectly, have a substantial positive effect, through organizational agility, on company performance. The depth and breadth of cloud deployment and organizational agility are also positively related. Furthermore, the complementary fit strategy has a major impact, whereas the balanced fit strategy has an insignificant effect on company performance.

**Keywords:** assimilation depth; assimilation breadth; organizational agility; firm performance; balanced fit; complementary fit





## 1. Introduction

Cloud computing is a core computing paradigm, with substantial benefits for information technology (IT)-related innovations coming in the future [1]. It offers convenient global remote use of shared IT resources, like storage and applications, servers, and networks, configured easily on demand without interaction from intermediaries like cloud platforms [2]. The use of cloud computing improves organizational agility in developing new products and services, expanding flexibility, increasing productivity, and building a sustainable partner relationship [3]. The conventional business model is transformed with cloud computing to encourage better collaboration, enabling effectively offered product lifecycle management and aligning product and service innovations with the business [4].

While it is believed that cloud computing provides multiple advantages for an enterprise, academic and industrial analysts remain concerned about the continuous, slow, time-consuming process and many instances of failure to recognize the real benefit of cloud computing [5–7]. Many firms are facing a challenge in crafting a strategy for executing and deploying cloud-based resources. A crucial reason is that firms focus mainly on the initial adoption of cloud computing in their business functions. However, early incorporation is only a stage of the entire process of assimilation, and only large-scale assimilation will truly accomplish the performance and value development of an IT [8,9]. Consequently, like other IT-based innovations for fostering the efficiency of business processes [9,10], cloud computing faces substantial assimilation gaps between the targeted adoption rates and actual use rates [11]. According to Zhu et al. [12], assimilation incorporates three stages

(e.g., evaluation, adoption, and deployment) of an innovation's complete life cycle, in which innovation is considered a vital element of the firm's value chain operations [9]. Thus, it is noteworthy to understand cloud computing assimilation's impact on firms' value creation and performance.

This study examines how firms configure cloud computing resources following the asset orchestration perspective. According to the asset orchestration perspective, firms may use their limited resources better by utilizing resource deployment techniques [13]. Resource deployment mechanisms are described by [14,15] as finding, choosing, configuring, and deploying limited resources and assets of enterprises. These deployment techniques can act as mediators between firms' assets and performance. Additionally, these strategies support managers' capacity to coordinate resources, convey vision, and encourage creativity [15]. Although extant literature has documented IT assimilation's overall effect on organizational-level value and performance based on IT [16–18], organizations are tangled in choosing the dimensions of cloud computing assimilation, such as the depth of assimilation and the breadth of assimilation, because these dimensions contest for scant resources [19]. According to organizational ambidexterity theory, firms should have the capability of ensuring a mix of both assimilation dimensions to improve firms' performance [20,21]. In addition, some studies from the fit perspective recommend alignment among various assimilation strategies, contexts, and structures to determine IT-enabled value [17,22,23]. Therefore, to understand how cloud computing improves firm performance, in-depth fit analysis is imperative to discover the combined impacts of different mutually dependent dimensions of cloud assimilation [11,17,24,25].

Moreover, some empirical studies have argued that the assimilation of IT does not influence the firm's performance directly; rather, some other capabilities work as mediators between the relationship of these two [26]. Felipe et al. [27] claimed that the impacts of information systems' (IS) capabilities on firm performance are mediated by organizational agility. Likewise, Sambamurthy et al. [28] revealed that the two crucial mediators between the relationship of IT capabilities and firm performance are knowledge management and agility.

Building on the above discussion, the authors aim to investigate the following research question (RQ): What is the impact of the depth and breadth of cloud computing assimilation on firm performance? In order to find the answer to RQ, the authors examine how several variables (such as the depth and breadth of cloud computing assimilation) work together. More specifically, this study attempts to answer the following sub-questions: (i) What are the discrete effects of a particular depth and breadth of cloud computing assimilation on company performance in accordance with IT assimilation theory [18] and IT value theory [29]? (ii) What are the combined impacts between depth and breadth of cloud assimilation on firm performance according to organizational ambidexterity theory [30]? (iii) What is the mediation effect of organizational agility between dimensions of cloud computing assimilation and firm performance? and (iv) What are the most critical factors that need special managerial attention to improve firm performance? An empirical investigation was conducted to examine the model and hypotheses based on a sample from Chinese firms. A systematic structure equation modeling—importance performance map analysis (SEM-IPMA) approach was applied to evaluate the research questions. The strength in path analysis of SEM was applied to investigate the first three questions, while the fourth question was assessed using IPMA. This study expands our knowledge of how the depth and breadth of cloud assimilation both separately and jointly contributes to firm performance. It also offers managerial implications for how firms can improve cloud-supported performance by appropriately integrating different cloud assimilation dimensions.

The article continues by describing the theoretical context. Next, this study builds the research model and hypotheses based on a thorough evaluation of the literature. The research approach is then illustrated. The study's data analysis, discussion, and implications are then given. Finally, the authors discuss the conclusion, the study's limitations, and potential future research areas.

## 2. Theoretical Foundation

### 2.1. IT Assimilation and Cloud Computing Assimilation

Since the inception of IT innovation diffusion, IS research has turned into an extensively widespread topic. An innovation includes a new idea, product, or process to an adopter [31]. Adoption of a new system by an organization demands some vital changes in its activities [32]. Diffusion, conversely, refers to the "process by which an innovation is communicated through certain channels over a period of time among the members of a social system" [31].

The nature of diffusion regarding IT innovation in firms is identified as a six-stage process of initiation, adoption, adaptation, acceptance, routinization, and infusion [33]. Additionally, Rogers [31] mentioned innovation diffusion as a sequential process of knowledge acquisition, persuasion, decision, implementation, and confirmation. The innovation diffusion follows a three-phase process: initiation, adoption, and post-adoption. The adoption decision denotes the acquisition of an IT innovation, and the deployment of IT innovation indicates the routinization of the innovation for regular use [31]. The implementation of an IT innovation involves high complexity; therefore, according to IT assimilation theory, there exists an assimilation gap for each IT innovation [12,34]. Subsequently, a firm cannot observe the value and the success of an IT innovation without its effective deployment [11,18,35,36].

For this, a number of scholars have started to devote themselves to conceptualizing IT innovation assimilation and its lifecycle [11]. Some researchers claim that IT assimilation comprises the entire route of IT diffusion [12,34]. They highlight that assimilation refers to the degree to which a firm proceeds through an innovation deployment phase, starting from initial adoption to routinization [9]. On the other hand, many researchers focus mostly on the post-adoption stage because of the pragmatic and theoretical significance of IT implementation. They signify that IT assimilation is analogous to IT integration [37] or IT alignment [38], which denotes the degree to which the usage of innovation infuses firms' operations and processes [35]. In this study, consistent with extant literature [9,16,18], cloud computing assimilation is defined as the extent to which the real utilization of cloud computing is harmonized with the firm's business processes or activities.

The utilization of cloud computing can be categorized into two particular dimensions, depth and breadth [17,39]. Depth describes the scale and intensity of cloud utilization. On the other hand, breadth indicates the scope and diversity of cloud utilization. Following this distinction, we explore the depth and breadth of cloud computing utilization as fundamental building blocks for deployment strategies. Firms with in-depth cloud computing usage frequently collaborate with external partners, while firms focusing on broad usage frequently promote relationships between firms [17].

From the process viewpoint of the IT value creation life cycle, IT invention is a facilitator of IT business value creation. [40]. Subsequently, cloud assimilation has become indispensable for value chain activities [9]. Therefore, after cloud computing adoption decisions, firms should stimulate the integration of cloud services and their business activities to ensure effective assimilation of cloud computing for enhancing performance.

### 2.2. IT Business Value and Firm Performance

In the existing IS literature, IT value has been widely investigated. IT impacts organizational performance through profitability, productivity enhancement, cost reduction, inventory reduction, and competitive advantage [29]. Numerous researchers have identified the IT business value's concept and dimensions based on myriad approaches [17,29,41]. When firms utilize IT-enabled cloud services, they can improve financial and operational performances [3,5]. Operational performance supported by IT results in enhancements in the efficiency of the business process. In contrast, financial performance results in an improvement in the firm's overall profitability, equity, market share, or assets [27,42,43].

Compared with its rivals, the primary aim of managing an organization is to achieve superior performance. Performance evaluation is undoubtedly regarded as one of the

common and recurring subjects in the management literature [27]. There is, however, a debate about the concept, dimension, and output measurement of performance [44]. Scholars have calculated the financial return of businesses from two viewpoints in the literature: financial and non-financial [45,46]. Financial success depends not only on the company's effectiveness but also on the sector where the company operates. [27]. For this reason, not all financial ratios are equally reliable and significant when measuring an organization's financial results.

Enterprises' efficiency can be evaluated by applying two methods: objective and subjective. In this analysis, subjective measures have been applied to determine the firm efficiency of enterprises, since the inclusion of non-financial subjective indicators provides a clearer view of the operating factors behind financial success [47]. Subjective metrics are built with respondents' expectations of how well their business does [27,43].

### 2.3. Organizational Agility

Organizational agility is the firm's unique capability, which aids in addressing changes and uncertain events in firms' surrounding business environment and responding quickly to those recognized changes and uncertain events to capture prospects of expansion [48]. In the literature, three interrelated dimensions of organizational agility are intensively addressed: agility in operation, customers, and partners [28].

Operational agility means the magnitude to which organizations can effortlessly and rapidly redesign their operation to cope with the changing environment [49]. Customer agility allows firms to utilize market changes by offering new value propositions through internal operations adjustments to ensure customer retention [50]. Partnering agility refers to the efficient utilization of existing resources and relationships with stakeholders like contract manufacturers, suppliers, logistic providers, and distributors through alliances and partnerships [27]. It also allows a firm to build and maintain a comprehensive enterprise network to access those resources [28].

These three interrelated dimensions reinforce one other, and the higher capabilities from these dimensions are executed by agile organizations [51]. The dynamic and turbulent nature of the surroundings makes business environments not anticipatable. The capacity of organizational agility supports recognition of the growth of demand from inside and outside, crafting of new action plans for facing changes, and attaining desired goals [52]. Cloud technology deployment, done in a proper way, can offer organizations swift accessibility to IT resources, ensuring a smooth and fast way of service, which subsequently enhances the firm's organizational agility [53].

### 2.4. Organizational Ambidexterity

Cloud computing assimilation's impact on firm performance requires cogitating the combined effects of different dimensions of assimilation. Existing literature proposes that the organizational ambidexterity theory [54] can be applied to examine the alignment strategies among diverse dimensions of cloud computing assimilation and their causal relationships with firm performance.

The word 'ambidexterity' originated from the Latin word 'ambos,' which denotes the capability of humans who can use both hands skillfully [21,55]. In organizational management research, ambidexterity is a metaphor for firms adroit at exploring and exploiting [20]. March [54] first introduced 'exploration and exploitation' as a twin concept. Various terms like variation, flexibility, experimentation, discovery, risk-taking, play, and innovation characterize the concept of exploration [54]. Exploration underlines fundamental innovation actions that suffer from high risk in exploring novel knowledge and resources, and these actions are noteworthy for sustainable improvement [56]. Exploitation is characterized by various activities, such as production, efficiency, refinement, choice, selection, and implementation [54]. Although both exploration and exploitation are crucial for a firm's survival and competitiveness [57], several researchers have specified an alignment challenge between exploration and exploitation, as these two contest for scarce resources, plus

they require dissimilar organizational procedures [17,54,56]. Accordingly, organizations that can both explore and exploit and improve the organizations' performance and ensure long-term survival [11] are named 'ambidextrous organizations' [21].

Prior studies have given attention to the conceptualization, antecedents, consequences, and measurement of organizational ambidexterity, considering the significance of organizational ambidexterity. Moreover, researchers have applied organizational ambidexterity to assess many organizational issues, such as strategic organizational management, alliances, technology innovation management, and learning [11,17,20,21,57]. Additionally, ambidexterity's implications are obvious in numerous recommendations offered for the improvement of organizational performance [54,58].

Precisely, Cao et al. [20] identified two dimensions of ambidexterity: the balanced dimension (BD) and the combined dimension (CD). The BD resembles an organization's focus on a suitable balance in exploration and exploitation [54], while the CD indicates their joint magnitude [58]. These two dimensions have a significant causal interaction to foster firm performance and ensure long-term survival [17].

## 3. Conceptual Framework and Research Hypotheses

We investigated how IT business value creation is influenced by the impact mechanism of IT assimilation and examined how various assimilation strategies based on IT can influence firm performance. Following the definition of IT assimilation dimensions [17], we divided cloud computing assimilation into two dimensions, depth and breadth. According to the IT business value theory, we considered firm performance as a matter of financial and operational performance [5,27]. According to previous research on IT value and IT assimilation [16,17,34], this study posits that computing assimilation's depth and breadth may influence the performance of a firm. According to the concepts of IT assimilation and innovativeness [59,60], we also propose that these two dimensions of assimilation improve a firm's performance by enhancing its agility. According to Felipe et al. [27], organizational agility is well accepted as the most significant higher-order capability that reveals strong, sustainable organizational performance, irrespective of the organizations' size and industries. Additionally, following organizational ambidexterity theory [20,56], we restate the significance of the strategic fit between cloud deployment's depth and breadth. Thus, we assume that cloud-supported organizational performance is directly influenced by the balanced fit approach and complementary fit approach. Finally, we have considered four control variables, specifically, firm size, annual revenue, industry, and firm age, to lessen the inconsistency in firm performance. The proposed research model is depicted in the following Figure 1.

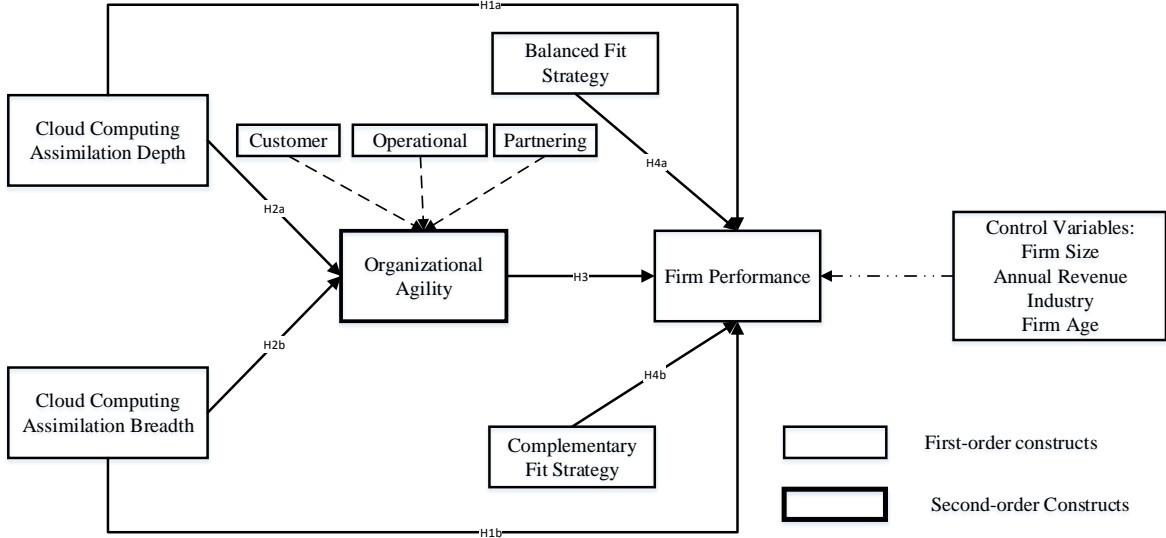

**Figure 1.** Conceptual model.

### 3.1. Cloud Computing Assimilation and Firm Performance

Prior literature suggests that IT usage and assimilation enhance organizational performance in the private and public sectors. Klein [16] found that internet-based purchasing application assimilation is positively related to operational performance. Similarly, Zhang et al. [17] identified that IOS deployment has a significant positive impact on firms' operational performance. Liang et al. [11] revealed that cloud assimilation in e-Government can improve both operational and strategic value in China's context. Cloud computing's unique characteristics empower firms to lift limitations on massive cash outflows for IT infrastructure, software, and equipment. Additionally, firms can enjoy mass IT resources at a lower cost with higher efficiency [11]. Thus, this study states that effectively realizing cloud computing assimilation may enhance IT business value.

In an organization, the assimilation of cloud computing follows a narrow-to-deep process. Initially, this process includes the usage of common cloud facilities like storage, then it gradually spreads to more complex services [61]. Therefore, the two dimensions of cloud computing assimilation are revealed, depth and breadth.

When the depth of cloud computing deployment is enhanced, the extent of cloud services usage for supporting core business is significantly greater. The advanced level of cloudization allows business expansion by minimizing production and transaction [62]. Additionally, more cloudization helps the firm improve organizational capabilities that enable intra-organizational and inter-organizational information sharing, cooperation, and collaboration [6]. In the breadth of cloud computing assimilation, when the firm spreads the breadth of cloud computing deployment, the diversity and number of agencies and services are to be increased. The breadth of cloud computing assimilation enables firms to diminish the cost of IT via pay-per-use-based services. Additionally, the breadth of cloud assimilation spreads digitization and ensures data sharing among different departments [11]. Therefore, the following hypotheses are proposed in this study:

**H1a.** *The depth of cloud computing assimilation positively influences firm performance.*

**H1b.** *The breadth of cloud computing assimilation positively affects performance.*

### 3.2. Cloud Computing Assimilation and Organizational Agility

The proper assimilation of IT is essential for organizational agility development, improving a firm's capacity to identify and react to a changing environment [63,64]. With cloud computing, business process can be upgraded to support digital options and innovation [3], subsequently improving organizational agility [28]. Nowadays, organizations are busy handling vast amounts of collected information from internal operations and the external environment. This large-scale information analysis is needed to ensure prompt responses to the challenges faced by firms. The perfect deployment of cloud computing supports business agility as well as IT efficiency [65]. Cloud-computing-supported business agility requires the quick deployment of mass IT, directly minimizing the cost of capital and rapid response to the market [65]. With digitization in operation and production in this age of Industry 4.0, cloud computing offers higher organizational agility [3].

More specifically, the depth of cloud computing assimilation enables a firm to improve its operational agility by signifying the need to detect changes, threats, and opportunities and offer fast and accurate responses to stakeholders by redesigning internal processes [66,67]. In contrast, the breadth of cloud computing assimilation helps it establish and maintain a good relationship with its customers and partners by offering new and innovative goods and services and strengthening external networks with its suppliers, contract manufacturers, distributors, and logistic supporters [68]. In sum, cloud technology implementation can ensure optimum organizational agility by addressing environmental changes, increasing innovativeness, scanning information from internal operation, and initiating quick internal operations changes. Thus, we hypothesize the following.

**H2a.** *The depth of cloud computing assimilation has a positive effect on organizational agility.*

**H2b.** *The breadth of cloud computing assimilation has a positive effect on organizational agility.*

*3.3. Organizational Agility and Firm Performance*

In the continuously changing environment, organizational agility assists performance growth through value capture and value creation. Organizational agility helps a firm identify, capture, and improve capabilities, which cumulatively support it in facing challenges introduced by different catalysts, such as competitors, suppliers, customers, and technology change [69]. Sambamurthy et al. [28] stated that firm performance could be increased through organizational agility because it fosters innovations in process, product, channels, relation, and marketplace segmentation. Strong customer agility supports firms in initiating prompt and proactive responses toward customers' demand by ensuring innovative offers in products, services, and promotions, which increase the revenue and competitive advantage of the firms [70]. Similarly, operational agility plays a significant role in improving a firm's competitive performance by ensuring operational flexibility, customer retention, and cost reduction [71]. Finally, firm performance is affected by partnering agility in strategic network development and trusted relationship improvement with partners using virtual platforms [28]. According to the above discussion, we posited the following hypothesis.

**H3.** *Organizational agility has a positive effect on firm performance.*

*3.4. Strategic Alignment between Cloud Computing Assimilation*

Despite the various benefits of utilizing IT/IS, many organizations fail to gain real IT value [72,73]. Hence, many researchers have suggested using an appropriate fit strategy to study the combined impacts of various interdependent IS adoption strategies [11,17,24].

In the cloud computing context, firms with appropriate cloud assimilation in both the depth and the breadth dimensions are possibly able to outperform those that are more arbitrarily linked [74]. According to organizational ambidexterity theory [54], depth and breadth are two building blocks of value-creation and IT-assimilation mechanisms in the firm; the asset orchestration view recommends that the combined effects of depth and breadth impact firm performance instead of their single independent effects [11,17,20].

Organizational ambidexterity suggests an effective fit among various strategies is essential for reaping organizational value [54]. According to Venkatraman [75], there are six fit perspectives, such as fit as moderation (FMO), fit as matching (FMA), fit as profile deviation, fit as mediation, fit as gestalts, and fit as covariation. In organizational ambidexterity studies, FMO and FMA are frequently applied. For instance, prior literature utilized the balance dimension (BD) and the combined dimension (CD) of ambidexterity [20]. The BD signifies that the balanced and consistent nature of the two dimensions is vital for the higher joint effect of these dimensions. The CD underlines the complementarity of these two dimensions and specifies that both dimensions can be reinforced together for a marginal effect on organizational performance [20].

Therefore, this study concentrates on these two categories of fit (e.g., balanced fit and complementary fit) between the depth and breadth of cloud implementation. A balanced fit denotes a firm's strategy that equally emphasizes both the depth and breadth of cloud computing deployment. In contrast, complementary fit refers to a firm's strategy to emphasize either depth or breadth to enhance the additional impact of these two dimensions [11].

For a better understanding of the concepts of these fit strategies, the subsequent mathematical formula is used. Similar to Cao et al. [20] and Zhang et al. [17], suppose a firm assigns a percentage of X of resources to enhance the cloud computing assimilation depth and a percentage of Y to enhance the cloud computing assimilation breadth. Both balanced fit and complementary fit define how cloud assimilation depth and cloud assimilation breadth together influence organizational performance. The term $1 - |X - Y|$ is used to express the performance effect of balanced fit. The lesser deviation between X and Y signifies the more considerable degree of balance between X and Y. Conversely, the term X*Y is used to express the performance effect of complementary fit, since a rise in the value

of X or Y may possibly increase the marginal effect of the other [17]. The larger the product of X and Y is, the greater the level of complementarity between X and Y [11].

### 3.5. Balanced Fit Strategy and Firm Performance

As the combined effect of the dimensions of assimilation (e.g., depth and breadth) on firm performance is more complicated than the individual direct effects of either dimension [17], this study further investigates how both assimilation dimensions of cloud computing could combinedly affect the firm performance by differentiating between the two fit strategies.

Both dimensions of cloud computing assimilation, depth and breadth, are similarly significant for enhancing the firm's performance. However, these two dimensions compete for scarce resources [17,40]. An inequity between these two dimensions creates a threat to firm performance because of subsequent risks. In contrast, when placing too much concentration on the depth of cloud assimilation, although this is able to provide more deep-level applications such as data analysis for the firm, the other systems' development will be lagged behind. This issue may lead to a 'staggered informatization level' problem in various firm processes, which is unfavorable to enhancing the total performance of cloud-driven operations and processes [11]. In contrast, by paying more attention to the breadth of assimilation, cloud computing can reduce IS's maintenance and operation costs and improve operational efficiency by utilizing resources [76]. However, there is a possibility of having a lack of deeper-level applications such as decision support.

In addition, Chang et al. [77] reveal that cloud computing supports a balanced IT ecosystem that enhances firms' cloud computing absorptive capacity. Such absorptive capacity allows firms to face challenges at the assimilation stage of cloud computing [57]. Accordingly, a balanced fit between these two dimensions enables firms to utilize cloud computing services to their fullest for enhancing firm performance. Therefore, we posit the following hypothesis.

**H4a.** *Balanced fit strategy positively affects firm performance.*

### 3.6. Complementary Fit Strategy and Firm Performance

Along with the balanced fit, the complementary fit is one more possible approach through which the depth and breadth of cloud computing may combinedly increase the performance of a firm, although there is the unbalanced assimilation [17]. The key concept of complementary fit states that cloud assimilation's depth and breadth complement each other to reinforce their marginal impacts on the performance of a firm [17,20]. Prior literature identified that IT's integration of various inter-firm coordinations has a promising ability to offer a competitive advantage [78,79].

From the cloud computing perspective, the complementary fit strategy stresses the complementary relationship between the two dimensions, the depth and breadth of cloud computing assimilation, plus it exploits the marginal impact on firm performance [20]. This complementarity approach can encourage different enterprise systems' transformation to cloud computing and enhance cumulative organizational performance [11]. Therefore, we formulated the following hypothesis.

**H4b.** *Complementary fit strategy positively affects firm performance.*

## 4. Research Methodology

A variance-based partial least-squares structural equation modeling (PLS-SEM) approach was applied to assess the research model because (i) PLS-SEM can examine a group of relationships simultaneously [80], (ii) the research has been designed for theory exploration or theory development instead of theory confirmation [3], (iii) the research model contains a large number of indicators that make the model more complicated, and (iv) the study mainly focuses on increasing the variance explained instead of focusing on the estimation of model fit [3]. Moreover, data distribution, sample size, single-item

constructs, and the study's exploratory nature make PLS-SEM the most suitable approach. In this study, SmartPLS software has been used to test the research model.

### 4.1. Constructing a Survey Instrument

We have constructed a survey instrument to gather empirical data. All the constructs were retrieved from the literature and adapted according to our study context. Appendix A shows the measurement items of the constructs and their literature sources. A five-point Likert-scale, ranging from 1 as 'very low' to 5 as 'very high', was used for all the main constructs' measures, except control variables. By following the prior literature, we measured the control variables [5,17,46]. Additionally, we measured the impact of balanced fit as $1 - |depth - breadth|$ and complementary fit as depth*breadth, and we standardized the scores of these depths and breadths for the calculation [11,17]. Therefore, assimilation depth and breadth have single-item measures.

The measurement instrument was pre-tested before distributing among Chinese firms to check the appropriateness of the format and wording. There are two sections in the questionnaire; section A contains demographic information about respondents and represented firms. This section includes respondents' occupational position, associated industry, age of the firm, annual revenue, employee size, and scope of the business. Section B contains questions measuring the constructs used in the model. First, the questionnaire was developed in English. After that, with the help of a professional translator, it was translated into the Chinese language. Then, this Chinese questionnaire was translated into English again. Two experts evaluated this back-translated instrument with excellent command of English and knowledge related to the research context. Based on experts' comments, modifications were made regarding the items' structure, content, and wording.

### 4.2. Data Collection Procedure

We obtained data from 296 Chinese firms, which have experience in utilizing cloud computing in their operations and processes. We requested from the firms that the questionnaire be answered by the top-level managers (e.g., CIO, senior IT/IS manager, etc.) involved with the cloud computing assimilation process in their firms. We distributed survey instruments both electronically and on paper.

We followed a non-probability sampling, specifically, the approach of 'key informant', for data assortment to include only the survey participants responsible for cloud-computing-related projects and who have cloud computing knowledge [1,11]. A total of 1000 survey instruments were dispersed, and, among those, 312 questionnaires were returned from the respondents, counted as a 31.2% rate of response. Thus, after data cleansing, 296 instruments were used for further analysis. The demographics of the sample are expressed in Table 1. The manufacturing industry (39.19%) dominated the participation of sampled firms, while the service industry represented 36.15%, and the trading sector depicted 24.66%. The majority of participating firms have had business operations for at least twenty years. Besides, 18.92% of the firms had from 500 to 1000 employees. In terms of annual revenue, the dominant group had 500 to 1000 million (32.77%).

**Table 1.** Demographic profile of the sample.

| Descriptions | | Frequency | Percentage |
|---|---|---|---|
| | Lower level | 52 | 17.57% |
| Managerial position | Mid-level | 146 | 49.32% |
| | Top level | 98 | 33.11% |

**Table 1.** *Cont.*

| Descriptions | | Frequency | Percentage |
|---|---|---|---|
| Work experience (Years) | 0 to 2 | 18 | 6.08% |
| | 3 to 5 | 37 | 12.50% |
| | 6 to 10 | 94 | 31.76% |
| | More than 10 | 147 | 49.66% |
| Firm age (Years) | Less than 5 | 12 | 4.05% |
| | 5 to 10 | 28 | 9.46% |
| | 10 to 15 | 42 | 14.19% |
| | 15 to 20 | 85 | 28.72% |
| | More than 20 | 129 | 43.58% |
| Employees | Less than 100 | 14 | 4.73% |
| | 100 to 500 | 29 | 9.80% |
| | 500 to 1000 | 56 | 18.92% |
| | 1000 to 1500 | 54 | 18.24% |
| | 1500 to 2000 | 45 | 15.20% |
| | More than 2000 | 98 | 33.11% |
| Annual sales (million $) | Less than 100 | 28 | 9.46% |
| | 100 to 500 | 45 | 15.20% |
| | 500 to 1000 | 97 | 32.77% |
| | 1000 to 1500 | 43 | 14.53% |
| | More than 1500 | 83 | 28.04% |
| Industry type | Manufacturing | 116 | 39.19% |
| | Service | 107 | 36.15% |
| | Trading | 73 | 24.66% |

*4.3. Assessment of Multicollinearity*

We checked if there was a multicollinearity issue with the data. The correlation matrix table, presented in Table 2, showed that the values for all the coefficients are less than the recommended value 0.90 [81], and constructs' VIF scores are below the cutoff 10 [82]. Hence, we did not find any multicollinearity problem among the constructs.

**Table 2.** Discriminant validity and variance inflation factor (VIF).

| Constructs | BF | CA | CF | DEP | FP | OPA | PA | BRE | VIF |
|---|---|---|---|---|---|---|---|---|---|
| BF | 1.000 | | | | | | | | 1.406 |
| CA | 0.070 | 0.874 | | | | | | | 1.740 |
| CF | 0.361 | −0.293 | 1.000 | | | | | | 1.492 |
| DEP | 0.212 | 0.557 | −0.349 | 0.891 | | | | | 1.889 |
| FP | 0.105 | 0.676 | −0.224 | 0.644 | 0.768 | | | | |
| OPA | 0.003 | 0.447 | −0.258 | 0.343 | 0.536 | 0.826 | | | 1.473 |
| PA | 0.048 | 0.540 | −0.251 | 0.551 | 0.676 | 0.496 | 0.846 | | 1.842 |
| BRE | 0.237 | 0.383 | −0.085 | 0.380 | 0.534 | 0.400 | 0.424 | 0.857 | 1.409 |

*4.4. Common Method Bias (CMB)*

According to the suggestions of Schwarz, Rizzuto [83], the problem of control common method bias (CMB) was acknowledged and controlled during the design stage of this study. For this, both procedural remedies and statistical tests of CMB were applied. To ensure

procedural remedies, we ensured respondents that their individual identities and responses would not be disclosed according to the recommendations of Ooi, Lee [3]. Moreover, we encouraged the participants to provide unambiguous responses and special attention to the wording of questions to decrease unequivocal responses. In the statistical tests for addressing CMB, we used Harman's single-factor approach. Principal axis factor analysis was applied to detect critical factors for the variance explained [84]. Only 28.05% of the total variance is explained by the most powerful single construct, which is less than the threshold of 50% [85]. Further, in the correlation matrix (see Table 2), no correlation was found among the constructs of more than 0.90 [86]. Furthermore, we tested CMB by using the partialling out of general factor or common factor method. By following this approach, when we added the general or common factor to the model, we noticed that the $R^2$ value for the endogenous constructs was not significantly increased (<0.10), which reduces the concern about the CMB problem [85,87]. Values of variance inflation factors (VIF) of the constructs (see Table 2) were also used to verify CMB, and the values were found below the recommended cutoff value of 3.3 [88]. Hence, there is no serious concern about a CMB problem in this study.

## 5. Results

### 5.1. Measurement Model

We examined the internal reliability, convergent validity, and discriminant validity of the variables used in the model [80]. Three popular indicators of assessing constructs' reliability are Cronbach's alpha, Dijkstra–Henseler's rho (ρA), and composite reliability. The criteria values of reliability were found larger than the recommended value of 0.70 [89] (see Table 3). Additionally, item loadings were used to measure the reliability of items and all the loadings were found larger than the suggested value of 0.70. We measured convergent validity by average variance extracted (AVE), and its scores for AVE are more than the recommended score of 0.50 [90]. Besides, the composite reliability scores were greater than the suggested value of 0.70 [89]. Similarly, in Table 2, discriminant validity was inspected by identifying the correlations among the constructs of probably overlying variables. The square root of AVE for each individual construct was found to be correlated to a greater extent with other constructs than itself. For double confirmation, all the HTMT ratios (see Table 4) were less than 0.90. Thus, there is evidence of the discriminant validity of all study constructs [91].

**Table 3.** Measurement model.

| Constructs | Items | Scale Type | Loadings/Weights [a] | t-Value | Cronbach's Alpha | rho_A | CR [b] | AVE [c] |
|---|---|---|---|---|---|---|---|---|
| First-order constructs | | | | | | | | |
| Assimilation depth | DEP1 | Reflective | 0.906 | 80.949 ** | 0.871 | 0.873 | 0.921 | 0.795 |
| | DEP2 | | 0.879 | 51.259 ** | | | | |
| | DEP3 | | 0.889 | 67.566 ** | | | | |
| Assimilation breadth | BRE1 | Reflective | 0.870 | 75.289 ** | 0.828 | 0.888 | 0.893 | 0.735 |
| | BRE2 | | 0.885 | 48.434 ** | | | | |
| | BRE3 | | 0.816 | 27.594 ** | | | | |
| Firm performance | FP1 | Reflective | 0.766 | 30.350 ** | 0.901 | 0.903 | 0.920 | 0.591 |
| | FP2 | | 0.794 | 30.635 ** | | | | |
| | FP3 | | 0.777 | 33.085 ** | | | | |
| | FP4 | | 0.784 | 32.680 ** | | | | |
| | FP5 | | 0.760 | 28.040 ** | | | | |
| | FP6 | | 0.718 | 23.775 ** | | | | |
| | FP7 | | 0.734 | 27.890 ** | | | | |
| | FP8 | | 0.812 | 44.246 ** | | | | |
| Balanced fit | BAL | Reflective | 1 | N/A | 1.000 | 1.000 | 1.000 | 1.000 |
| Complementary fit | COM | Reflective | 1 | N/A | 1.000 | 1.000 | 1.000 | 1.000 |
| Second-order constructs | | | | | | | | |
| Operational agility (OPA) | OP1 | Formative | 0.413 | 20.862 ** | N/A | N/A | N/A | N/A |
| | OP2 | | 0.414 | 19.942 ** | | | | |
| | OP3 | | 0.385 | 22.194 ** | | | | |

**Table 3.** *Cont.*

| Constructs | Items | Scale Type | Loadings/Weights [a] | t-Value | Cronbach's Alpha | rho_A | CR [b] | AVE [c] |
|---|---|---|---|---|---|---|---|---|
| Customer agility (CUSA) | CUS1 | Formative | 0.381 | 39.324 ** | N/A | N/A | N/A | N/A |
| | CUS2 | | 0.394 | 35.125 ** | | | | |
| | CUS3 | | 0.369 | 27.781 ** | | | | |
| Partnering agility (PARTA) | PART1 | Formative | 0.290 | 32.148 ** | N/A | N/A | N/A | N/A |
| | PART2 | | 0.308 | 37.983 ** | | | | |
| | PART3 | | 0.291 | 32.613 ** | | | | |
| | PART4 | | 0.294 | 33.926 ** | | | | |

** significant at $p < 0.001$; [a] Loadings of reflective items and weights of formative items; [b] CR = Composite reliability, [c] AVE = Average variance extracted.

**Table 4.** Heterotrait–Monotrait (HTMT) ratio.

| Constructs | BF | CA | CF | DEP | FP | OPA | PA | BRE |
|---|---|---|---|---|---|---|---|---|
| BF | | | | | | | | |
| CA | 0.077 | | | | | | | |
| CF | 0.361 | 0.316 | | | | | | |
| DEP | 0.228 | 0.648 | 0.374 | | | | | |
| FP | 0.112 | 0.775 | 0.236 | 0.726 | | | | |
| OPA | 0.041 | 0.556 | 0.294 | 0.419 | 0.641 | | | |
| PA | 0.087 | 0.630 | 0.269 | 0.634 | 0.761 | 0.606 | | |
| BRE | 0.265 | 0.423 | 0.091 | 0.440 | 0.585 | 0.463 | 0.479 | |

Based on the recommended approach by Petter et al. [92], we evaluated the quality of second-order constructs, which are formative in nature. Initially, we assessed multicollinearity using the VIF of the formative items of organizational agility. We confirmed no multicollinearity with the evidence that all the values of VIF were less than 10 (see Table 5). Moreover, we measured all the indicators' weights for organizational agility, which were enormously significant (see Table 3). Lastly, the first-order constructs are significant as the indicators of the second-order constructs (see Table 6).

**Table 5.** Test of Multicollinearity.

| | Coefficients [a] | | |
|---|---|---|---|
| | | **Collinearity Statistics** | |
| **Model** | | **Tolerance** | **VIF** |
| 1 | CA | 0.666 | 1.502 |
| | OPA | 0.708 | 1.412 |
| | PA | 0.627 | 1.595 |

[a]. Dependent Variable: OA.

**Table 6.** Weights of first-order constructs towards second-order constructs.

| **Second-Order Construct/Path** | **Weight** | **t-Value** |
|---|---|---|
| Organizational agility | | |
| CA -> OA | 0.394 | 18.440 |
| OPA -> OA | 0.305 | 13.088 |
| PA -> OA | 0.517 | 23.438 |

*5.2. Structural Model*

Structural model analysis was employed to test the hypothesized relationships between constructs. The hypotheses results are depicted in Figure 2. The model explained 68.1% of the total variation in firm performance through the depth of cloud computing assimilation ($\beta = 0.285$, $p < 0.01$), breadth of cloud computing assimilation ($\beta = 0.175$, $p < 0.01$), organizational agility ($\beta = 0.553$, $p < 0.01$), and complementary fit strategy ($\beta = 0.090$, $p < 0.05$), which were statistically significant, but the balanced fit strategy

($\beta = -0.060$, $p > 0.05$) was not; respectively, hypotheses H1a, H1b, H3, and H4b are confirmed, but H4a is not.

**Figure 2.** Structural model. NOTE: * $p < 0.05$; ** $p < 0.01$, ns = not supported.

The model explained 45.1% of the variance in organizational agility, and both constructs were statistically confirmed, namely, depth of cloud assimilation ($\beta = 0.494$, $p < 0.01$) and breadth of cloud assimilation ($\beta = 0.305$, $p < 0.01$). Therefore, hypotheses H2a and H2b are supported.

In our study, most of the control variables, except annual revenue, such as firm size, industry, and firm age, were found to have an insignificant role ($p > 0.05$) in explaining the variation in performance. In contrast, firm performance is significantly affected by annual revenue ($p < 0.05$).

### 5.3. Predictive Relevance and the Effect Size

Cohen [93] recommended that predictive relevance be determined as low, moderate, and large based on $Q^2$ values of 0.02, 0.15, and 0.35, respectively. In this study, all endogenous constructs have large predictive relevance, as the $Q^2$ values for all the endogenous constructs (see Table 7) are above 0.35. Correspondingly, the predictive capability of the endogenous constructs can be specified using $R^2$ values. The $R^2$ values for organizational agility and firm performance range from 0.451 to 0.681 (as depicted in Table 7). Therefore, there is evidence of the acceptable predictive capacity of these endogenous constructs. Additionally, we measured the effect size by applying the $f^2$ values for organizational agility on firm performance. The $f^2$ value (>0.35) in Table 7 indicated a large effect size of organizational agility on firm performance.

**Table 7.** Predictive relevance and the effect size.

| Endogenous Constructs | $R^2$ | $Q^2$ | $f^2$ |
| --- | --- | --- | --- |
| Organizational agility | 0.451 | 0.428 | 0.504 |
| Firm performance | 0.681 | 0.648 | |

### 5.4. Mediating Effect

Baron and Kenny [94] state that a construct can be a mediator if it confirms some conditions: (1) the influence of the exogenous construct on the endogenous construct has to be statistically significant; (2) the effect of the exogenous construct on the mediator

construct has to be significant; (3) the effect of the mediator on the endogenous construct has to be significant; and (4) the exogenous construct must affect the endogenous construct less strongly with the presence of a mediator in the same model. However, the Baron and Kenny approach was criticized by Hayes [95] in his article "Beyond Baron and Kenny: Statistical mediation analysis in the new millennium". Later, Hayes [96] offered complete solutions to examine the effect of mediators by identifying the indirect effect's significance with the bootstrapping technique.

To examine the mediation effect of cloud computing assimilation's depth and breadth through organizational agility, we applied Preacher and Hayes' [97] approach as suggested by Hair et al. [80]. In this study, Preacher and Hayes [97] recommended that at 95% boot confidence interval, if the β coefficient at the upper limit [UL] and lower limit [LL] does not straddle a 0 in between, there is mediation. As depicted in Table 8, the results of the bootstrapping indicate that the indirect effects of the depth and breadth were significant. Thus, this study validates the mediation effect of organizational agility.

**Table 8.** Specific indirect (mediation) effect.

| Path | β | *t*-Value | *p*-Value | LL | UL | Mediation |
|------|------|-----------|-----------|------|------|-----------|
| BRE -> OA -> FP | 0.168 | 4.891 | 0.000 | 0.104 | 0.240 | YES |
| DEP -> OA -> FP | 0.273 | 7.901 | 0.000 | 0.199 | 0.337 | YES |

*5.5. Importance–Performance Map Analysis (IPMA)*

According to Ringle and Sarstedt [98], "The IPMA gives researchers the opportunity to enrich their PLS-SEM analysis and, thereby, gain additional results and findings. More specifically, instead of only analyzing the path coefficients (i.e., the importance dimension), the IPMA also considers the average value of the latent variables and their indicators (i.e., performance dimension)". The IPMA is being run to differentiate factors with comparatively low performance but with comparatively high importance (high total effect) in determining endogenous variables [80]. Therefore, the IPMA supports identifying areas with poor performance but with superior importance to a target construct [99], thus suggesting key areas to improve or emphasize.

This study performed importance–performance map analysis (IPMA) in PLS, considering firm performance as the target construct. According to the map (see Figure 3), we can observe that assimilation depth, organizational agility, and assimilation breadth have comparatively higher importance values (total effects) relative to other factors. Additionally, these constructs have higher importance scores as well. If we carefully observe, we can say that these three constructs are located in the map's upper right quadrant. This indicates that firms are managing these factors quite efficiently. However, the balanced fit strategy has a relatively higher performance score despite the lower importance score. Conversely, the complementary fit strategy has a higher importance score than the balanced fit strategy, although it has a lower performance score. Therefore, managers need to pay special attention to improving the performance of the complementary fit strategy while reducing the importance of a balanced fit strategy to enhance the firm's performance. Managers need to assign more resources in the complementary fit strategy, reassigning some resources from the balanced fit strategy.

It can be observed from Figure 4 that customer agility, operational agility, and partnering agility are the three dimensions of organizational agility, positioned within the upper right quadrant of IPMA graph. This result suggests special monitoring of these agilities, because the scores of these three agility dimensions are above the mean not only in terms of performance but also in terms of importance. Among them, partnering agility and customer agility seems to be the most significant dimensions, as the scores of both dimensions are well above the mean score of importance. Additionally, the score of operational agility is near to but above the mean of importance. Interestingly, there are no

dimensions that appear within the lower right quadrant, signifying that they are not critical areas for improvement.

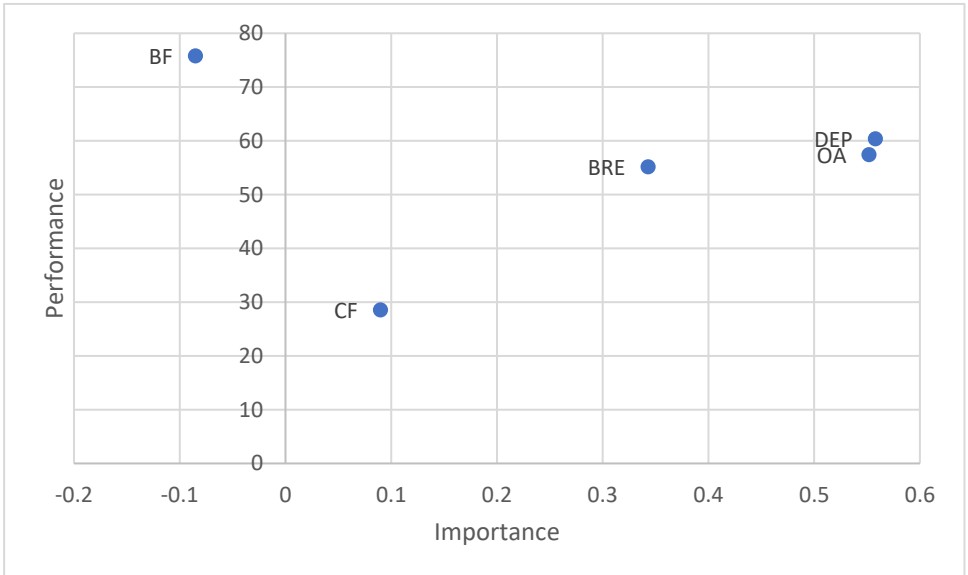

**Figure 3.** IPMA graph.

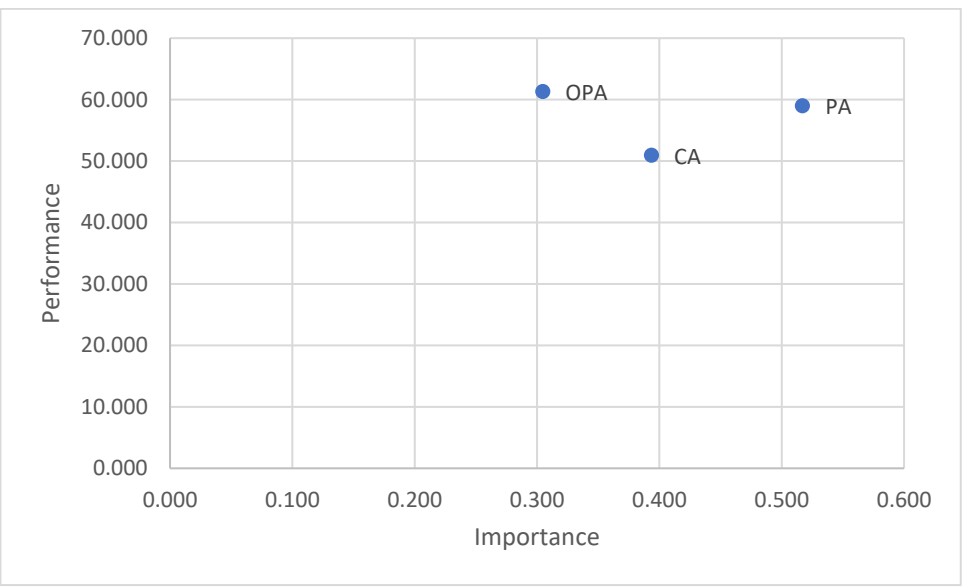

**Figure 4.** IPMA graph.

## 6. Discussion and Implications

This study considers the asset orchestration perspective [13] to examine alternative configurative approaches used to relate how firms deploy IS assets in cloud computing [17]. Consistent with several prior studies [11,16–18], this study identified that the depth and breadth of cloud computing assimilation have an essential effect on firms' performance. The basic dimensions of cloud computing usage reveal the degree and scope of the firm's real practice of cloud-driven services. According to previous literature, both dimensions enable firms to redesign conventional processes and operations with mass cloud computing capability and advanced information-sharing capacity [1,76]. The effective deployment of cloud computing technology facilitates innovation in the traditional business model, helping firms optimize business processes efficiently and integrate scattered information,

enhancing firm performance. Cloud capability helps firms increase non-financial performance through enhancements of the quality of products or services, corporate growth and market share, and financial performance, by expanding sales, revenue, profit margin, ROI, ROE, and ROA. This outcome is consistent with extant cloud computing literature [3,5]. However, the effects vary between two dimensions; for example, the impact of cloud assimilation depth on firm performance ($\beta$ = 0.285) is larger than the influence of cloud assimilation breadth ($\beta$ = 0.175).

Moreover, in our study, the effects of cloud computing assimilation's depth and breadth on organizational agility are found to be positively related, and organizational agility has a positive and strong influence on firm performance. This empirical study has demonstrated that organizational agility intervenes in the impact of the depth and breadth of cloud computing assimilation on firm performance. Many prior studies in different IT assimilation contexts support this observation [27,28,100].

This study also identifies the strategic fit between the dimensions of both depth and breadth of cloud computing deployment as antecedents of firm performance. Many studies on the orchestration perspective [13] and organizational ambidexterity [54] support this finding, which emphasizes that the synergy among IT strategies is essential for the improvement of organizational performance [79].

In particular, similar to Liang et al. [11], this study observes a strong positive connection between complementary fit strategy and firm performance. Although we know that the assimilation of cloud computing can enhance firm performance, there are still some challenges. For instance, although breadth of assimilation has the ability, to some extent, to enhance informatization level, minimize IT costs, and ensure long-term development, without depth of assimilation, which effectively ensures business process integration and optimization, it is tough to reap these benefits [16] ultimately. Depth of assimilation can help understanding of the effective management of a particular business through sensible business process optimization. According to Liang et al. [11], without the expansion of the depth application from a particular domain to other business domains, it is not possible to achieve inter-departmental synergy and data sharing, such as cloud silos. Thus, the complementary approach could compensate for the insufficiency of a particular dimension to a certain degree based on the firm's real situation.

On the other hand, a balanced fit strategy is not found to be an important determinant of firm performance. Although this finding is different from some earlier studies [17,20], this finding is supported by the discovery of Liang et al. [11]. The resource-based view (RBV) suggests that the limited resources' innovative utilization regulates an organization's competitive advantage [5]. According to the balanced fit strategy, firms make the investment in the same resources in depth and breadth, intending to make these dimensions reach an equal level. Practically, firms often suffer from resource limitations; therefore, it is problematic to invest enormous resources specifically to achieve a balance between them. Additionally, with balanced investment in the depth and breadth of cloud computing implementation, firms struggle to build their core competencies. In sum, according to the conflictual viewpoint of ambidexterity theory, the strategy of a balanced fit is not an active promoter. Thus, assimilation in a balanced way between these two dimensions cannot be justified from the performance improvement perspective.

## 7. Implications of the Research

### 7.1. Theoretical Implications

Our paper offers some mentionable contributions to the literature on cloud computing diffusion. In the IS literature, most of the prior studies have concentrated on the pre-stage of the adoption of innovative technology, such as on advantages and challenges, motivators and barriers, and predictors of cloud computing adoption by organizations. However, recently, scholars have suggested emphasizing the post-stage of the adoption of cloud computing diffusion [1,101]. Therefore, this study integrates IT business value, IT assimilation, and organizational ambidexterity theories to offer holistic insights into

the impacts of cloud computing assimilation on firm performance. We claim that cloud computing assimilation's depth and breadth affect firm performance in both individual and combined ways, recommending that cloud computing assimilation is a complicated process and needs innovative knowledge to be learned.

In the literature, some studies investigate the effect of IT assimilation in the organization [16–18], in which the solo impact of IT assimilation is emphasized. However, in response to the scholars' recent call, this study applies a comprehensive analytical approach to consider the match between cloud computing assimilation's depth and breadth and firm performance. This study examines both the distinct and the combined impacts of assimilation's depth and breadth on firm performance. This study then advises that a complementary fit strategy is much more essential for realizing actual benefits (firm performance) from cloud computing deployment.

In addition, this article underlines the significance of agility in creating business value for firms. The assessment of agility's contribution to the firm's value-creation mechanism has become a potential area of research [27]. One of the innovative contributions of this study is to build and confirm a model that implants organizational agility as a mediator between cloud computing assimilation and firm performance. This emergent research stream involves complex multi-layered interactions among various factors, requiring more investigation and empirical research [27]. This study offers holistic thoughts regarding the underlying interactions among IT assimilation, agility, and firm performance. This study confirms that the two fundamental dimensions of cloud computing implementation influence firm performance by incorporating the mediation effect of organizational agility. This study also conceptualizes agility as an upper-order capability, which is facilitated by cloud computing assimilation to achieve firm performance through incessant detecting and responding to a dynamic environment.

Finally, this research has executed importance–performance map analysis (IPMA) in PLS-SEM. An empirical basis was offered by the IPMA results to improve target constructs' performance [98]. According to the findings of IPMA, the depth and breadth of cloud computing assimilation as well as organizational agility are managed and controlled efficiently. On the other hand, more attention is needed in a complementary fit strategy to upturn its mechanism towards firm performance, as this strategy is identified with a lower performance score but a higher importance score. Additionally, IPMA outcomes signify that the most significant dimensions of organizational agility are partnering agility and customer agility, which suggests a careful control of these two dimensions by the managers.

*7.2. Practical Implications*

Cloud computing assimilation at the organization level involves a complicated and systematic venture, including technological and organizational restructure. The study outcomes can be valuable for designing and planning the firms' deployment and implementation and implementation of cloud computing for the managers.

The business value of cloud computing is a lengthy commitment; outcomes will not appear overnight. Therefore, firms need a long-term commitment of resources. As firms have a limited budget, it is difficult for the firm to invest more along the two dimensions of assimilation, depth and breadth. For this, top management should have a better understanding of the joint effect of cloud assimilation's depth and breadth in order to devise the strategy. This study suggests that the complementary fit strategy can be an effective way to improve cloud computing assimilation's overall business value.

On the one hand, to stimulate the depth of cloud computing, firms should utilize breadth. For instance, firms should reflect the principles of 'increment first', 'commonality first and special later', and 'easiness first and difficulty later', for the development of cloud computing [11]. According to the 'increment first' principle, firms directly deploy and implement in the cloud computing platform. According to the 'easiness first and difficulty later' principle, firms should first consider applications that are relatively easy to deploy and maintain, and, later, firms migrate to complicated systems. According to the 'common

first and special later' principle, firms first migrate to common types of cloud applications and then migrate to special applications. In sum, in line with the techniques described above, firms have to endeavor best to enhance cloud computing application scope (breadth), and, therefore, encourage the omnidirectional penetration (depth) of cloud applications in various business activities [11].

In contrast, the depth of cloud computing assimilation can be promoted through cloud applications' breadth. Firms can stimulate and support in-depth cloud applications on core business operations and processes, for example, using big data mining applications to solve complex business problems to enhance the business value of cloud computing. After experience with the in-depth application, the cloud application's scope can be spread to other operations and processes of the firm.

Finally, managers need to be conscious of the significance of enhancing organizational agility, which assists firms in identifying and exploiting business opportunities. Organizational agility also enables firms to be innovative to stay ahead of their competitors and optimize their performance [27]. Organizational agility influences firm performance through the constant reconfiguring of resources and capabilities. Therefore, managers in an organization are advised to recognize and evaluate how cloud computing assimilation can be embedded in fundamental business processes to facilitate an agile response.

## 8. Conclusions, Limitations and Future Research Directions

### 8.1. Conclusions

The performance impact of various cloud computing deployment alignment strategies on the operations and procedures of the company is investigated in this study. This article develops a research model of cloud computing assimilation and evaluates the individual and combined effects of the depth and breadth dimensions of cloud assimilation on firm performance on the basis of various underlying theories, such as information technology (IT) assimilation, IT value, and organizational ambidexterity. This study takes into account two deployment alignment strategies—the balanced fit strategy and complementary fit alignment—according to the asset orchestration perspective. These two strategies represent strategic decisions made by businesses regarding the emphasis they place on the depth and breadth of cloud technology deployment in their operations and processes. The data are analyzed using a hybrid methodology (PLS-IPMA). This study asserts that a firm's performance is impacted by the depth and breadth of its absorption of cloud computing. The alignment between the depth and breadth of cloud computing assimilation in a company's processes and operations must also be taken into consideration. According to our research, there is a considerable impact on business performance from the complementary fit approach between the depth and breadth of cloud computing assimilation. This finding will undoubtedly provide guidance when allocating resources to help businesses adopt cloud computing.

### 8.2. Limitations and Future Research Directions

Although the researchers followed a meticulous approach to validate the research model, the outcomes may be confined to a particular study setting. Firstly, the model was confirmed using a sample taken from China, and the findings reveal only the scenario of China. Thus, a further investigation is proposed with the data of other similar countries. Secondly, we gathered this study's empirical data from the firms of the urban area and ignored the firms of rural areas, which are not facilitated with sophisticated, cutting-edge technologies. Therefore, a future study can be conducted based on a larger sample of urban and rural areas of the country. Thirdly, as the research was done using cross-sectional single-point data, a further study using longitudinal data is suggested to validate this study's model. Finally, only linear statistical techniques were employed for analyzing the data, which demands a future study involving non-compensatory and nonlinear techniques like neural network and fuzzy set analysis.

**Author Contributions:** Writing—original draft, A.K.; Writing—review & editing, M.T.I.; Supervision, Y.B. All authors have read and agreed to the published version of the manuscript.

**Funding:** This research received no external funding. The APC was funded by the University of Dhaka.

**Institutional Review Board Statement:** Not applicable.

**Informed Consent Statement:** Not applicable.

**Data Availability Statement:** Not applicable.

**Conflicts of Interest:** The authors declare no conflict of interest.

## Appendix A

**Table A1.** Measurement items.

| Constructs | Items | Sources |
|---|---|---|
| Assimilation depth | DEP1. The degree to which cloud computing supports related basic business operations at your firm.<br>DEP2. The degree to which cloud supports related operational activities at your firm.<br>DEP3. The degree to which cloud supports related decision making at firm. | [11,17] |
| Assimilation breadth | BRE1. The types of cloud service adopted by your firm.<br>BRE2. The quantity of cloud service adopted by your firm.<br>BRE3. The quantity of systems migrated by your firm. | [11,17] |
| Operational Agility | OPA1. We have the ability to fulfill demands for rapid response.<br>OPA2. We can quickly increase or decrease our production/service levels to meet market demand fluctuations.<br>OPA3. If we face any supply chain interruption, we can make essential substitute measures and internal adjustments quickly. | [27,50] |
| Customer Agility | CUSA1. In the face of market/customer shifts, we are quick to take and execute the correct decisions.<br>CUSA2. We are constantly looking for ways to redesign and reengineer our organization to provide improved service to the marketplace.<br>CUSA3. We view market changes and apparent uncertainty as opportunities for rapid capitalization. | [27,50] |
| Partnering Agility | PARTA1. We collect detailed information about our suppliers and service providers.<br>PARTA2. We are able to exploit the resources and capabilities of suppliers to enhance the quality and quantity of products and services.<br>PARTA3. We work with external suppliers to create high-value products and services.<br>PARTA4. We are able to manage relationships with outsourcing partners. | [27,102] |
| Firm Performance | Compared with key competitors, our company.<br>FP1. Is more successful.<br>FP2. Has a greater market share.<br>FP3. Is growing faster.<br>FP4. Is more profitable.<br>FP5. Is more innovative.<br>FP6. Is more productive.<br>FP7. Has a greater operational performance.<br>FP8. Has more growth in sales | [5,27,46] |

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
