# Peer review of "Understanding the Effects of Alignments between the Depth and Breadth of Cloud Computing Assimilation on Firm Performance: The Role of Organizational Agility"

_sustainability, doi:10.3390/su15032412_

Round 1
Reviewer 1 Report
Dear Authors,
This paper is well written and provides an interesting empirical verification.
However, the paper is out of scope for Sustainability. In fact, there are no items in your empirical analysis that refer to sustainability aspects.
You should probably target other journals based on management or operations management with digitalization on cloud computing.
Best wishes.
Author Response
Comments:
1. This paper is well written and provides an interesting empirical verification.
Response: The authors would like to thank the reviewer for his nice words about our work.
2. However, the paper is out of scope for Sustainability. In fact, there are no items in your empirical analysis that refer to sustainability aspects. You should probably target other journals based on management or operations management with digitalization on cloud computing
Response: Thank you very much for your suggestions. We reviewed this journal's aim and scope before submitting the manuscript. However, as we have assessed the impact of cloud computing assimilation on firm performance, which is an indicator of sustainability, we can claim that this paper is within the journal's scope. Moreover, we found similar kinds of papers that were published in this journal. Anyway, thank you again for your nice and constructive words.
Reviewer 2 Report
Dear authors,
I have carefully read your article, and I think it is original and makes a contribution to the field.
I have only a few minor comments/suggestions:
- where you cite the source in the text, you wrote the first two surnames of the authors, e.g. p. 2, line 47. I think you should write the last name of the first author et al. or with co-authors. You have the same thing several times in the text. Check how it is specified in the article preparation instructions.
- p. 2, line 53 - delete information technology and leave only IT, since the abbreviation was defined earlier (the same applies to line 64); and on page 7, line 262
- p. 3, line 85 - missing space before [29]
- in some places in the text you have two spaces together (eg p. 3, line 93, line 106, line 118...).
- page 5, line 201 - there can't be the figure after the title, but at least some text and then a figure, so I suggest moving it to the end of the paragraph of this chapter
- page 8, line 344 - I'm not familiar with the word »reorces« - is this mistake?
- page 9 - you have to write which program you use for processing - SmartPLS, Warp (?) - you must also indicate the source of the program.
- I suggest that in the research methodology chapter, somewhere at the beginning of the chapter, you explain the steps of the rest of the chapter.
Kind regards,
Reviewer
Author Response
Comments:
1. I have carefully read your article, and I think it is original and makes a contribution to the field.
Response: The authors would like to thank the reviewer for his kind comment about our manuscript.
2. I have only a few minor comments/suggestions:
Response: Thank you very much for your suggestions. We have made the necessary corrections incorporating all of your valuable suggestions in our revised manuscript.
Reviewer 3 Report
Thanks to select me to review the paper. There are something need to review before publication.
1. Abstract needs to review.
2. The introduction: I could not identify the research gap of the paper. Please add main issue with sub issues, idetify the research objectives with 3/4 researh questions. Then, stated the research gap.
3. In the implications of the study: Identify the novelty of the research and how your research is contributing the industry/society.
4. Write the conclusion.
Thank you
Author Response
Comments:
1. Abstract needs to review.
Response: Thank you for your suggestion. We have reviewed the abstract and adjusted it accordingly.
2. The introduction: I could not identify the research gap of the paper. Please add main issue with sub issues, idetify the research objectives with 3/4 researh questions. Then, stated the research gap.
Response: Thank you very much for your constructive suggestions. We have substantially improved our indroduction section based on your given suggestions. We have stated the research gap and research questions clearly in our revised manuscript. Please see the introduction section of the revised version of our manuscript.
3. In the implications of the study: Identify the novelty of the research and how your research is contributing the industry/society.
Response: In our revised manuscript, there is a chapter named implications of the research (Chapter 7). In this chapter, there are two sections; one is theoretical implications and the other is practical implications.
4. Write the conclusion.
Response: Thank you. In our revised manuscript, we have added a conclusion.
Reviewer 4 Report
The manuscript at this state is very interesting both for academia and professional. I just have two questions : how could the authors define agility in the case study section? And how a company could implement this methodology efficiently in their process? This second question will conduct the authors to show the balanced performance before and after the implementation.
Author Response
Comments:
The manuscript at this state is very interesting both for academia and professional. I just have two questions : how could the authors define agility in the case study section? And how a company could implement this methodology efficiently in their process? This second question will conduct the authors to show the balanced performance before and after the implementation.
Response: Thank you for your nice and constructive words.
Q1: how could the authors define agility in the case study section?
Answer: We have conducted. Organizational agility is the firm's unique capability, which aids in addressing changes and uncertain events in firms’ surrounding business environment and responding quickly to those recognized changes and uncertain events for capturing prospects of expansion [48]. Based on an extensive literature review on organizational agility, In our study, we have incorporated three interrelated dimensions of organizational agility: agilities in operation, customers, and partners [28]. Then we developed a formative construct. Details are mentioned in the manuscript
Q2: This second question will conduct the authors to show the balanced performance before and after the implementation
Answer: We obtained data from firms with cloud computing experience in their operations and processes. We requested the firms that the questionnaire be answered by the managers involved with the cloud computing assimilation process in their firms. Basically, we have conducted our research based on their responses. There is no comparison of performance before and after implementation. Maybe this could be an interesting agenda for future research.
Round 2
Reviewer 1 Report
After the revisions, I still believe that this paper is not in line with Sustainability. I then leave the decision to the Editor, also considering a plagiarism report of 18%.
